# Structure and density of silicon carbide to 1.5 TPa and implications for extrasolar planets

D. Kim [1✉], R. F. Smith[2], I. K. Ocampo[1], F. Coppari [2], M. C. Marshall [3], M. K. Ginnane [3], J. K. Wicks[4], S. J. Tracy[5], M. Millot [2], A. Lazicki [2], J. R. Rygg[3], J. H. Eggert[2] & T. S. Duffy[1]

There has been considerable recent interest in the high-pressure behavior of silicon carbide, a potential major constituent of carbon-rich exoplanets. In this work, the atomic-level structure of SiC was determined through in situ X-ray diffraction under laser-driven ramp compression up to 1.5 TPa; stresses more than seven times greater than previous static and shock data. Here we show that the B1-type structure persists over this stress range and we have constrained its equation of state (EOS). Using this data we have determined the first experimentally based mass-radius curves for a hypothetical pure SiC planet. Interior structure models are constructed for planets consisting of a SiC-rich mantle and iron-rich core. Carbide planets are found to be ~10% less dense than corresponding terrestrial planets.

[1] Department of Geosciences, Princeton University, Princeton, NJ, USA. [2] Lawrence Livermore National Laboratory, Livermore, CA, USA. [3] Laboratory for Laser Energetics, University of Rochester, Rochester, NY, USA. [4] Department of Earth & Planetary Sciences, Johns Hopkins University, Baltimore, MD, USA. [5] Earth and Planets Laboratory, Carnegie Institution for Science, Washington, DC, USA. ✉email: donghoonkim86@gmail.com

A large number of extrasolar planets (>5000) have been discovered in recent years. The chemical compositions of their host stars are diverse, implying that these planets may exhibit a wider range of composition than found in our solar system[1–3]. Carbon planets are one of the possible types of exoplanets, proposed to form around host stars having a high carbon-to-oxygen ratios[4,5]. High C/O ratios above 0.8 may alter the condensation sequence in protoplanetary disks and result in the formation of planets with a mantle dominated by C, SiC, and other carbides rather than silicates[1,5,6]. Although conflicting results have been reported in the literature, recent measurements suggest that a small number of host stars may have high C/O ratios[7–11]. In particular, a recent survey[8] of the chemical composition of a large number of stars found that ~1% had C/O > 0.8 (for comparison, the C/O ratio of the Sun is 0.55[12]). In view of the very large number of planets expected to exist in our galaxy, these results imply there may exist many carbon-rich planets of various sizes. Planetary formation processes could further enhance formation of carbon-rich planets even for host-star C/O ratios as low as 0.65[13]. Moreover, carbon planets are of special interest even if found in only a small percentage of planets as they would represent a highly exotic planetary style that would likely have a very different surface environment, tectonic style, heat flow, and potential for habitability than typical silicate planets[14].

Although silicon carbide (SiC) is expected to be a major constituent of this novel planetary type, its properties at the very high pressures of exoplanetary interiors are poorly constrained. Shock-wave experiments have reported a phase transition in SiC with a large volume decrease (~20 %) near 100 GPa[15–17], but the data are restricted to less than 210 GPa[17]. Under static compression in a diamond anvil cell (DAC), the phase transformation from B3 to B1 was reported to occur ~60–70 GPa[18,19]. The equation of state of B1 SiC was determined up to 200 GPa[18,20]. However, the existing studies are in strong disagreement when the reported equations of state are extrapolated to the higher pressures relevant to exoplanets that are several times more massive than Earth.

Dynamic ramp compression is a technique to access lower than shock temperatures, in which materials can be compressed in the solid state to extreme pressures, avoiding the melting that occurs under shock loading[21]. By combining ramp-compression experiments with in situ X-ray diffraction (XRD), the atomic-level structure of materials can be determined under extreme conditions reaching into the terapascal regime (1 TPa = 10 million atmospheres)[21–25].

In this study, the structure and equation of state of SiC were examined using laser-driven ramp compression and in situ XRD up to ultrahigh pressures of 1.5 TPa. We present the first direct constraints on the structure and equation of state of SiC under dynamic loading at conditions relevant to the deep interior of extrasolar planets.

## Results

**Laser-driven ramp compression**. Ramp-compression experiments were conducted using the Omega-60 and Omega-EP lasers at the Laboratory for Laser Energetics (University of Rochester, NY). The high-intensity Omega lasers were used to both drive a ramp compression wave into the sample on ~6-10 nanosecond timescales, and to generate a ~1 or 2-ns duration quasi-monochromatic X-ray source. The X-rays were used to record diffraction patterns from the compressed sample shown in Figs. 1 and 2, and stresses were determined from interferometric wave profile measurements shown in Figs. 3 and 4 (see "Experimental methods" for details). The experimental data are summarized in Supplementary Material Table 1.

SiC was compressed to stresses ranging from 80(4) GPa to 1507(64) GPa. Figure 5 shows the observed lattice $d$-spacings measured by X-ray diffraction. The low-pressure zincblende-type (B3) phase was observed at 80(4) GPa, the lowest stress achieved in this work. This measured stress is higher than the reported B3-B1 transition stress (~60−70 GPa) from laser-heated diamond cell experiments[18–20] but below the transition stress observed in shock wave studies (~100 GPa)[15], indicating that kinetics and/or stress heterogeneity play a role in this transformation under dynamic loading. For shots between 140(5) GPa and 371(11) GPa, two or three diffraction peaks were observed that could be indexed as the (111), (002), and (022) reflections of the rocksalt-type B1 phase of SiC (Fig. 5). Thus, the phase transformation to the B1 phase occurs between 80 GPa to 140 GPa under ramp loading. These results are comparable to previous continuum gas-gun experiments[15], and in situ XRD results from laser-driven shock experiments, which observed the formation of B1 SiC at ~114 GPa[17]. At stresses between 433(15) and 621(20) GPa, only one diffraction peak from the sample was observed which could be indexed as the B1 (002) peak, the most intense reflection for this structure. (Fig. 5). The B1 (111) peak could not be observed in this stress range due to overlap with diffraction from the pinhole material (Supplementary Fig. 1). At stresses above 700 GPa, 1-2 diffraction peaks are again observed and these can be indexed to B1 (111) and (002). SiC was compressed to ~9.77(29) to 10.07(30) g/cm³ at peak stress, yielding a 3-fold increase over its ambient density value (Supplementary Fig. 3 and Table 2). There is no evidence for any further transformations[26] or dissociation of SiC (Supplementary Figs. 1 and 2). At stresses above 371 GPa, the increased scatter in densities is likely due to a reduction in signal to background ratio as the result of the high X-ray background emitted by the drive plasma for higher stress experiments (Fig. 5).

**Equation of state of B1 (rocksalt-type) SiC**. Ramp compression thermodynamically mimics a series of weak shocks, thereby reducing the excess heat arising from increased entropy during compression. However, irreversible work and heating as a result of dissipation and material strength occur during ramp loading, so the loading path is generally considered quasi-isentropic. We have applied a series of corrections to transform from the longitudinal stress inferred from the velocimetry measurements to the corresponding principal isentrope and 300-K isotherm (see "Experimental methods" for details). For our ramp-compressed B1 SiC data, the reduced isentropes and isotherms were fit to a third-order Birch-Murnaghan (BM) equation of state to place constraints on the corresponding bulk modulus, $K_0$, and its pressure derivative, $K_0'$. Considering the uncertainties in the thermodynamic and strength parameters of B1 SiC, however, the differences between reduced isentropes, isotherms and ramp-compressed EOS are small. Here we have assumed a principal isentrope for B1 SiC constructed assuming a yield strength, $Y = 20$ GPa, and a Gruneisen parameter, $\gamma_0 = 1$. The calculated energy and temperature along the Hugoniot, ramp and principal isentropic paths are shown in Supplementary Figures 6 and 7. The zero-pressure volume, $V_0$, of the B1 phase was fixed at 66.3 Å³ (Ref. [18]). The resultant fit parameters of the reduced isentrope are $K_0 = 286(20)$ GPa and $K_0' = 3.83(16)$. If we fit the isentrope to a Vinet equation of state, we obtain the following fit parameters: $K_0 = 265(26)$ GPa and $K_0' = 4.4(3)$ (Fig. 5 and Supplementary Table 3).

Our results are compared with extrapolations of previous 300-K static equations of state data (black and blue lines) and first-principles calculations at 0 K for B1 SiC (Fig. 5, Supplementary Table 3). The two previous static equations of state fits[18,20]

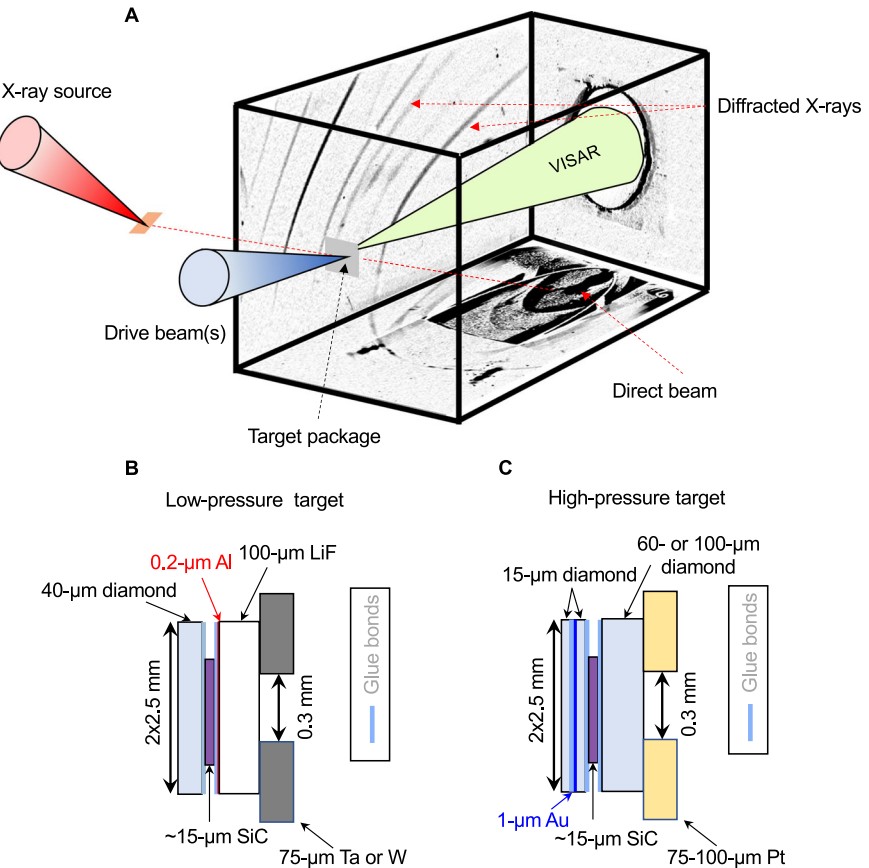

**Fig. 1 Experimental set-up and target packages for laser-driven compression. a** The drive laser (blue cone) generates stress waves through the C/SiC/ window target package (gray square). The stress rapidly equilibrates in the SiC foil following reverberation of the compression waves due to impedance mismatch with the diamond and window interfaces. A quasi-monochromatic X-ray source was generated by laser irradiation (red cone) of a Cu or Ge foil (orange square). The diffracted X-rays from the target are recorded on image plates inside of a detector box. Target assemblies used for low-stress (<400 GPa) and high-stress (>400 GPa) experiments are shown in **b** and **c**, respectively. The front diamond surface is illuminated by the drive laser using distributed phase plates (800- or 1100-μm focal spot). The incident X-rays are collimated using a 300-μm diameter pinhole (W, Ta, or Pt).

diverge when extrapolated above ~600 GPa. Our data are in good agreement with the extrapolation of the results of ref. [20] but not ref. [18] Theoretical calculations predict a wide range of stress-density relations for SiC[26–33] (Fig. 5, Supplementary Table 3), but it should be noted that our results are in agreement with the recent theoretical calculations of ref. [26] Thus, the results of this study together with those of Refs. [20] and [26] show good overall consistency in the equation of state determined from dynamic experiments, static experiments, and theory at ultrahigh pressures. A summary of experimentally constrained stress, $d$-spacings, densities and phase assignments for ramp-compressed SiC is shown in Supplementary Table 2.

## Discussion

Planetary accretion simulations suggest carbon-rich planets may be dominated by carbon-rich phases in the mantle and Fe-Ni-Si compounds in the core[1]. To evaluate the interiors of possible carbon-rich exoplanets, we constructed a simplified internal structure model for an eight-Earth-mass planet in which the mantle consists of SiC or a mixture of C (diamond) and SiC (1:1 molar fraction) and the core consists of Fe or an Fe-Si alloy with 15 weight % Si (hereafter Fe-15Si). By analogy with the Earth, the mantle is assumed to be 70% of the planet by mass and the core 30% by mass. The proportion of light elements in the core will depend on the partitioning behavior of iron and light elements during planetary formation processes[34,35]. Here we use ramp-compression data[23] for Fe-Si up to 1300 GPa as a proxy

for possible light elements and to minimize the need for extrapolation.

The interior structure was calculated using the BurnMan package[36] to solve coupled equations for hydrostatic equilibrium, mass conservation, and the equation of state of each layer given the fractional mass of the core and mantle. We neglect temperature effects which have been shown to have only a small effect on mass-radius relationships[37,38]. In contrast to previous studies, we included the B3-B1 phase transition at 70 GPa in our model which results in an increase in the planet's radius for a given mass. The calculation used the present results from B1 SiC together with previous experimental results for Fe-Si[23], Fe[39], and B3 SiC[18]. Carbon-rich planets are predicted to have a graphite-rich crust[14], which is not included in our model as the effect of the crust on the mass-radius relations is negligible. The surface environment and atmosphere composition of carbon-rich planets may be very different from silicate planets[14,40].

The resulting 8-$M_E$ planets have radii of 1.79-1.84 $R_E$, shown in Fig. 6, where $R_E$ and $M_E$ are the radius and mass of Earth, respectively. The pressure at the core mantle boundary is 0.9-1.1 TPa and the central pressure ranges from 2.7 to 3.4 TPa (Fig. 6). The presence of a light element in the core increases the core size but substantially decreases the central pressure. This is consistent with the previous models for silicate-dominated planets[23,38,41,42]. In the mantle, the phase transition from four-coordinated SiC (B3) to six-fold coordination (B1) occurs at a shallow depth ($R/R_P \simeq 0.93$, where $R_P$ is a planetary radius) and

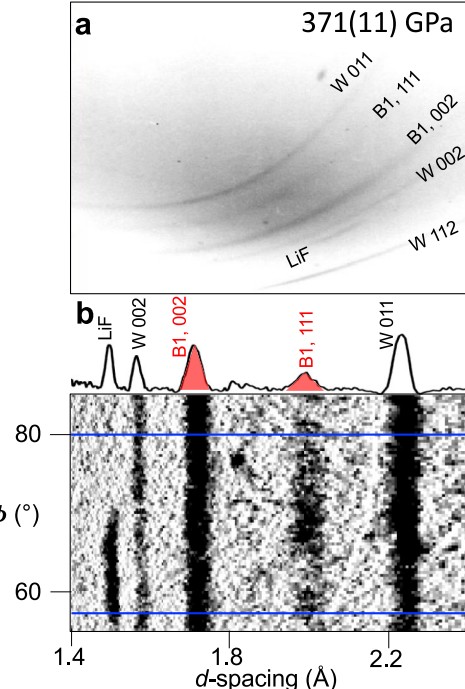

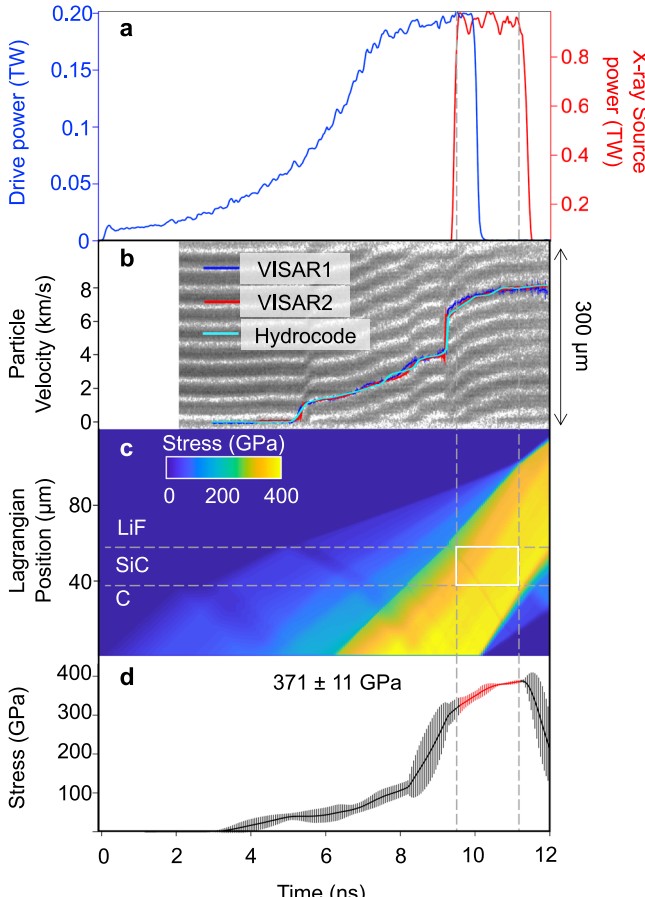

**Fig. 2 X-ray diffraction pattern for a representative experiment on SiC (shot #27430). a** One panel of an unprocessed image plate. **b** Projection of the above image plate into $d$-spacing-$\phi$ space after background subtraction. The one-dimensional X-ray diffraction pattern shown on the upper axis represents integration in $\phi$ over the region between the blue lines. The textured diffraction from LiF can be distinguished from the more extended features from either the W pinhole or B1 SiC.

**Fig. 3 Timing and stress determination for SiC ramp-compression experiment (LiF window). a** Drive laser pulse for shot #27430 (blue trace). X-rays are generated using a 2-ns square pulse (red trace). **b** Raw interferogram from the VISAR records the SiC-LiF interface particle velocity (solid lines) which is reproduced by a hydrocode simulation (light blue curve) to determine **c** stress history in the target package. The white rectangle represents the sample stress condition during the X-ray probe period (bounded by vertical dashed lines). The horizontal dashed lines represent the material layers in Lagrangian coordinates. **d** Calculated stress history of the SiC sample as a function of time. The vertical dashed lines represent stress states over the X-ray probe period.

the B1-phase SiC would comprise more than 77 vol.% of the 8-$M_E$ planet's mantle. In contrast, the mantle of an Earth-sized carbon planet would be mainly composed of the low-pressure B3 phase (~90 vol.%, $R/R_P \simeq 0.63$)[43]. The B3-B1 transformation is accompanied by a large density increase of 15–20%[15,18–20,44]. Such a significant density increase combined with a negative Clapeyron slope for the transition[18–20] may inhibit convection across the boundary, favoring a layered thermal convection pattern suggested by previous studies[43]. The large transition energy[27] may also favor layered convection. The reduced mass and heat transfer between the lower and upper mantle may affect the thermochemical evolution of the planet.

Incorporation of diamond into the mantle increases the radius of planets with the same mass up to 2% (Figs. 6 and 7). Furthermore, diamond is predicted to transform to the BC8 phase above 1 TPa accompanied by ~3% density increase and metallization[26,45], although the transformation has not yet been directly experimentally confirmed[21]. In our model the predicted transition occurs for planets greater than about eight Earth masses. The thickness of the BC8-containing layer above the core-mantle boundary would be ~800–1400 km for the largest planet size we considered (10 $M_E$). The presence of a dense, metallic layer above the core may affect heat flow, thermal evolution, and magnetic field generation.

SiC is known to be a refractory material with a high melting temperature[46]. Melting is an important process in planetary interiors and strongly influences differentiation and the early history of a planet. Mantle viscosity is also an important factor for the internal structure and dynamics of planets. The high melting temperature and possible high strength of SiC suggest that the interior may be characterized by high viscosity and hence sluggish convection[47] even for the more highly coordinated B1 phase. It

should be noted that there are only limited measurements of the strength of SiC under at extreme conditions[16]. Additionally, SiC also has a high thermal conductivity and low thermal expansivity, which may further suppress mantle convection[18,43,48]. This contrasts with recent studies[24,49] that have suggested that viscosity decreases at high pressure for large silicate planets due to phase transitions of MgO and the dissociation of MgSiO₃. Recent numerical simulations have reported that the onset of convection in a carbon planet may be highly dependent on the activation volume and the initial temperature[18]. More data is needed on the properties of the B1 phase of SiC and SiC + diamond mixtures at extreme conditions to better understand the dynamic behavior of such planets.

Observations of the mass and radius of exoplanets provide a basis for modeling the possible range of their mineralogies and interior structures. Accurate equations of state to ultrahigh pressures are required to constrain the internal composition of exoplanets. We constructed an experimentally constrained mass-radius curves for a hypothetical pure SiC planet compared with other hypothetical single-phase planets (e.g. $H_2O$, C, MgSiO₃, Fe-15Si, Fe) for evaluating the possible compositional space of

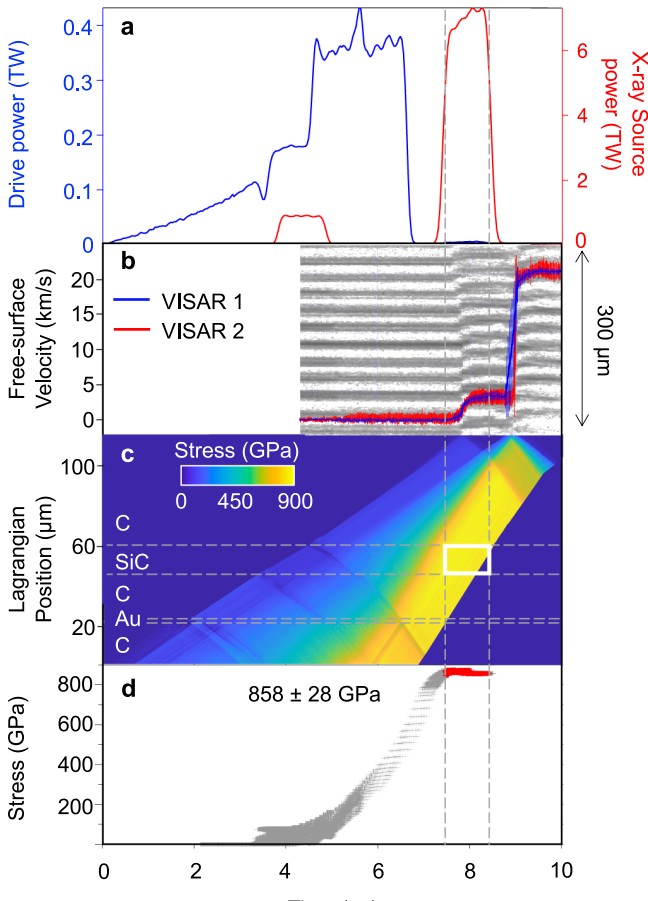

**Fig. 4 Timing and stress determination for SiC ramp-compression experiment (diamond window). a** Drive laser pulse for shot #98955 (blue trace). X-rays are generated using a pair of 1-ns square pulses (red trace). **b** Raw interferogram from VISAR records diamond free-surface velocity that is used to determine the stress history. **c** Calculated map of stress distribution within the target assembly as a function of time determined by the backward characteristics analysis (see Methods). The horizontal dashed lines represent the material layers in Lagrangian coordinates. **d** Calculated stress history of the SiC sample as a function of time. The vertical dashed lines represent stress states over the X-ray probe period.

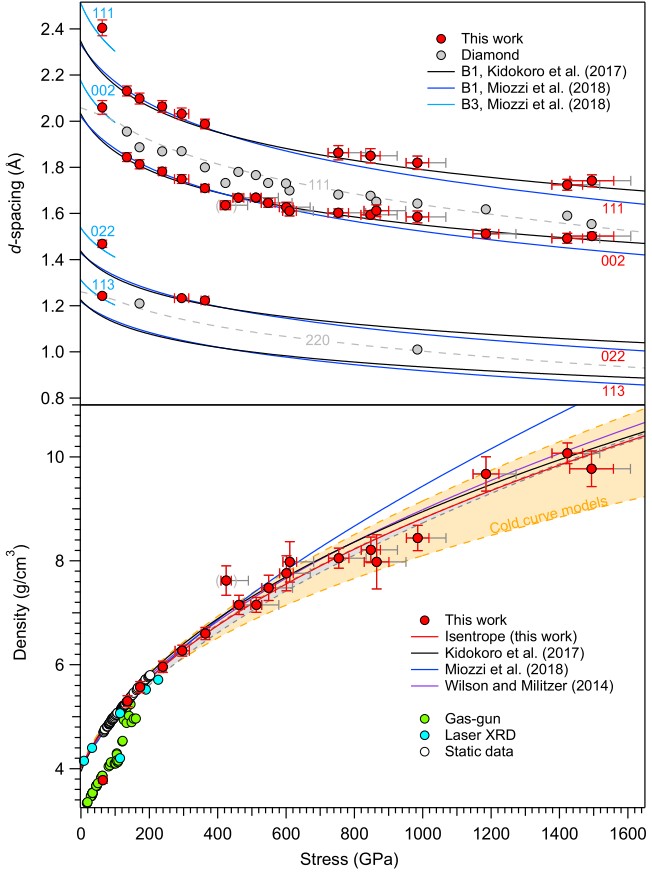

**Fig. 5 Measured diffraction peak positions and densities of SiC as a function of stress. a** The observed *d*-spacings of ramp-compressed SiC sample and diamond ablator/window are shown as red and gray circles, respectively. The solid lines are from static 300-K equation of state fits and their extrapolation as described below. The diamond ramp equation of state[64] and its extrapolation (>800 GPa) are shown as gray dashed curves. **b** Measured densities of ramp compressed SiC (red circles) and a Birch-Murnaghan (BM) equation of state fit to the reduced isentrope (red curve with assumption of Y = 20 GPa and $\gamma_0 = 1$) are compared to the extrapolated EOS of diamond anvil cell experiments (blue[20] and black[18] curves) and the range of densities obtained from first principles calculations (orange shaded region[20,26–33], see Supplementary Table 3). The gray band shows the pressure-density path along the principal isentrope calculated using the thermodynamic parameters that are listed in the Suppl. Table 4. Our EOS curve is consistent with the theoretical calculations of ref. [26] (purple curve). Previous shock compression data (cyan[17] and green circle[15,16]) and static diamond anvil cell data (open circles[18]) are also plotted. The datum at 433 GPa (gray parentheses) was excluded from fitting to the EOS (See Suppl. Table 2 for details). The 1σ error are shown as red bar. Gray error bars represent additional +50 GPa stress uncertainty due to uncertainty in the strength of diamond.

exoplanets. Our results show that a pure SiC planet is expected to be ~10% less dense than a MgSiO$_3$ planet. The mass-radius curves for a pure SiC are also compared with those of (yellow region) using the previously reported B1 EOS[18,20,26–33] shown in Fig. 7. These show a wide range of mass-radius curves due to long extrapolation of low-pressure data. Our study provides experimental data on the mass-radius relationship for a pure SiC planet to reduce uncertainties from extrapolation of the equation of state. Mass-radius curves are also calculated for a planet composed of a SiC mantle or a C and SiC (1:1 molar fraction) mantle (70% by mass) and an Fe core containing up to 15 weight % Si core (30% by mass). Figure 7 shows the mass-radius relationship for such planets extending up to 10 Earth masses in size. The red and gray regions in the figure represent the range of mass-radius curves depending on the core composition. Our result shows that SiC is less dense than MgSiO$_3$[50] and hence a planet with a SiC (red region) or C and SiC (gray region) mantle would be lighter than a corresponding Earth-like planet (30 % Fe + 70 % MgSiO$_3$) with the same radius[50]. For a given total mass, the radius of a planet decreases with core size (or mass) due to the high density

of the metallic core[42]. The mass-radius relation of carbon planets of 1–10 Earth masses in size would overlap those of Earth-like planets (30% core mass) if the core mass of the carbon planet is in the range of 38–49%.

Our results provide the first experimentally constrained mass-radius curves for SiC-rich bodies that do not rely on theoretical calculations or long extrapolation of low-pressure data. It should be noted that it is not possible to uniquely identify a planet's interior compositions based on the mass-radius relationship alone. Additional constraints such as detection of atmospheric contents and/or host star composition may help reduce

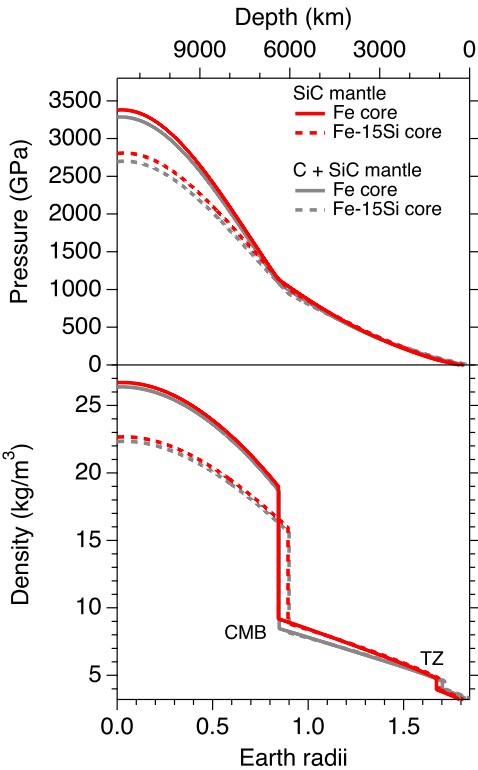

**Fig. 6 Interior pressure and density for an eight-Earth-mass carbide planet.** The carbide planet is modeled with a 30% by mass Fe or Fe-15wt% Si (Fe-15Si) core and a 70% SiC or C and SiC (1:1 molar fraction) mantle. The addition of Si in the core decreases pressures and densities in the core but increases core size. CMB, core-mantle boundary; TZ, transition zone (due to the B3 to B1 transition in SiC).

compositional degeneracy[4]. Spectroscopic measurements of exoplanetary atmospheres in future space missions may enable improved characterization of super-Earth planets and resolve the question of the existence and composition of carbon-rich planets.

Our results provide experimental evidence for the stability of SiC in the B1 phase over a wide range of exoplanetary conditions that encompass expected mantle conditions of carbon-rich exoplanets. The effect of other potential planetary constituents, if present in sufficient quantity, on phase relations and other properties has been explored only under limited conditions so far[51,52]. Further work is needed to better understand the mineralogy of carbon-rich rocky planets as a function of pressure, temperature, composition, and oxidation state.

In sum, the atomic-level structure of SiC has been determined under dynamic ramp loading to 1.5 TPa using in situ X-ray diffraction. Our data confirm that SiC transforms to the B1 structure between ~80 and 140 GPa and remains in this phase to at least 1.5 TPa without further phase transformation. Our work extends the experimental constraints on this material to extreme pressures, more than seven times greater than those achieved with standard static-compression techniques, and more than four times greater than the central pressure of the Earth. SiC was compressed to densities more than three times larger than its ambient value. Our results enable construction of experimentally constrained mass-radius curves for SiC-rich planets. Carbide planets are found to have a lower density than Earth-like planets. Interior structure models of carbide planets have also been constructed, placing constraints on the internal pressure and density distribution, and yielding insights into the possible dynamics and thermal behavior of such bodies.

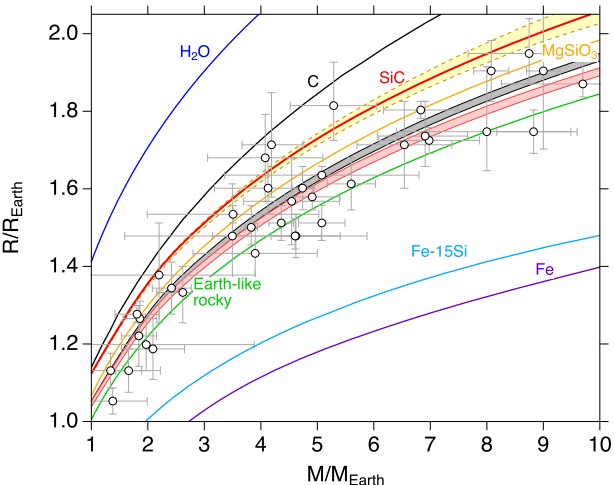

**Fig. 7 Calculated mass-radius relationships compared with observed exoplanets.** Mass-radius curves are shown for hypothetical planets composed entirely of SiC (this study), diamond[26], $H_2O$[50], $MgSiO_3$[50], Fe-15Si[23], and Fe[72]. The mass-radius relationship for an Earth-like rocky planet is shown in green. The yellow band represents the mass-radius relationship for a SiC planet using the B1 SiC equation of state from previous studies[18,20,26-33]. The pink band shows the predicted mass-radius relationship for a planet consisting of pure SiC mantle and a Fe or Fe-15Si core (mass fraction of core assumed to be 30%). The gray band shows the mass-radius relationship for a planet with 50% SiC-50% diamond mantle and a Fe or Fe-15Si core (mass fraction of core assumed to be 30%). Observed exoplanets are shown as gray circles with error bars (http://exoplanetarchive.ipac.caltech.edu). $R_E$ and $M_E$ are the radius and mass of Earth, respectively.

## Methods

**Sample preparation and target assembly.** Silicon carbide powder (~1-μm mean grain size, 99.8% purity from Alfa Aesar) was used as a starting material. The phase and purity of the sample (B3-type SiC, $F\bar{4}3m$) were confirmed by Raman spectroscopy. Sample disks of ~12−21-μm thickness were produced by compressing the SiC powder in a diamond anvil cell to 5 GPa, thereby reducing the porosity and increasing the uniformity of the sample. The chosen sample thickness range was a balance between maximizing diffraction intensity while avoiding the formation of shock waves in the sample. A representative compressed sample was examined by scanning electron microscopy. Identifying void space in the resulting image using the optical imaging method of ref. [53] leads to an estimated porosity of ~2% (Supplementary Figure 4).

The sample package was mounted on the front of a detector box lined with image plates (Fig. 1A). The target assembly consisted of a SiC layer sandwiched between a [110] diamond ablator and either a [100] LiF window for low-stress experiments (<400 GPa) or a [110] diamond window for higher stresses (>400 GPa) (Fig. 1B). A thin Au layer was added in the latter case to prevent preheating and melting of the sample by X-rays emitted by the drive ablation plasma. Epoxy layers were ~1-μm in thickness. A Ta, W or Pt pinhole (75 to 150-μm thick, 300-μm-diameter hole) was centered over the back of the target package.

**X-ray diffraction measurements.** Samples were compressed using a laser drive consisting of one or more 351-nm ramp-shaped laser pulses. Either a single 10-ns long pulse (Omega EP) or a composite drive comprising three to six 1-3-ns long laser pulses (Omegea-60) was used. The drive laser directly ablates the diamond at the front of the sample package to generate a rapidly expanding plasma that drives compression waves through the SiC sample. Reverberations due to impedance mismatch across the boundary between SiC and LiF or diamond window (Fig. 1) generate a uniform stress state within the SiC layer. The peak drive laser intensities ranged from $1 \times 10^{13}$ to $1.1 \times 10^{14}$ W/cm². 

A quasi-monochromatic $He_\alpha$ X-ray source was produced by laser-irradiating Cu ($He_\alpha = 8.368$ keV) or Ge ($He_\alpha = 10.249$ keV) foils (Fig. 1A)[54]. The foils were positioned 24 mm from the target at an angle of 22.5° (Omega-EP) or 45° (Omega-60) from target normal. At Omega-EP a single 1-2-ns duration square laser pulse was applied with an energy of 1250–1950 J/beam. At Omega-60, 16-18 beams with energies of 400–500 J/beam were applied over a 1-1.6 ns square pulse. The X-ray drive lasers were timed to produce X-rays at the predicted time of peak stress within the SiC layer. The X-rays were transmitted through the sample and collimated by the pinhole. X-ray diffraction data were recorded on five image plates attached to the inside of the detector box[22]. The image plates were filtered by Cu

(12.5-μm thick) or Al (50- to 75-μm thick) to reduce satellite emissions such as $H_\alpha$, $He_\beta$ and $He_\gamma$ X-rays, and the bremsstrahlung X-ray background from the drive plasma[54]. Black kapton filters (25-μm thick) were placed in front of the Cu or Al filters to block optical light. Diffraction from the pinhole substrate provided ambient reference lines to calibrate the diffraction geometry (see Supplementary Table 1). The image plates were projected into $2\theta$-$\phi$ space ($2\theta$ is the scattering angle and $\phi$ is the azimuthal angle around the incident X-ray direction). In these coordinates, diffraction data are projected as straight lines of constant $2\theta$ (Fig. 2). Interplanar $d$-spacings were determined from the diffraction angle using Bragg's Law: $\lambda = 2d\sin(\theta)$, where is $\lambda$ the X-ray wavelength.

For each image plate, the observed diffraction features can be assigned to one of the following components of the target assembly: (1) the pinhole substrate material, (2) compressed SiC, (3) Bragg or Laue diffraction from compressed LiF or diamond. We observed between one and four diffraction lines from SiC in these experiments. Diffraction from LiF or diamond could be distinguished based on $d$ spacings as well as characteristic textural features. A representative Debye-Scherrer X-ray diffraction pattern for a typical experiment is shown in Fig. 2. X-ray diffraction patterns and filtered image plates for all shots are shown in Supplementary Fig. 1.

Systematic corrections were applied to correct for the offset of the pinhole substrate from the pinhole center and the offset of the sample and the pinhole, as described in ref. [55] The uncertainties in the interplanar $d$-spacings include variation in the measured value of two-theta as a function of azimuthal angle ($\phi$), uncertainty in fitting the sample peak positions to Gaussian profiles, and the uncertainty in the incident X-ray wavelength (<0.01 Å)[54]. The resultant uncertainties in $d$-spacing (~1%) are given in Supplementary Table 2.

**Stress determination.** A line-imaging velocity interferometry system for any reflector (VISAR)[56] was used to determine the velocity at either the free surface of the rear diamond or at the interface between SiC and the LiF window. The VISAR records the change in the Doppler shift of light reflected off a moving surface (Figs. 3 and 4). Two independent VISAR channels with different velocity sensitivities were used to resolve velocity ambiguities that exceed the frequency response of the system. Example velocity profiles are shown in Figs. 3B and 4B. As the strength of SiC on ramp loading is poorly constrained, we have not made any correction for the difference between axial stress ($P_X$) and mean pressure ($P$). The stress within the sample is calculated based on the full-width at half-maximum of the stress histogram over the duration of the 1-2-ns X-ray measurement.

For low-stress shots (<400 GPa), the SiC-LiF interface particle velocity, $u_p(t)$, was recorded as a function of time (Fig. 3B, blue and red traces). A correction to the interface velocity was made using the density-dependence of the refractive index of LiF[57]. The stress history within the sample was determined using the radiation hydrodynamics code HYADES[58]. The simulations were used to constrain the stress history within the sample using EOS descriptions of each target material and to provide an initial estimate of the delay time between the laser used for ramp compression and those used to generate the $He_\alpha$ emission (Fig. 3). An initial estimate of the stress is determined from a scaling law[59] that relates laser intensity to diamond ablation stress: $\sigma(\text{GPa}) = 42(\pm 3)[I(\text{TW/cm}^2)]^{0.71(\pm 0.01)}$. The actual stress history within the sample following ramp compression was determined by matching the observed VISAR profile to the hydrodynamic simulations. The SESAME EOS tables[60] for LiF (#7271v3) and diamond (#7830) and the Livermore Equation of State (LEOS) table[61] for SiC (#2130) were used in hydrocode simulations with a Steinberg-Guinan strength model[62] (shear modulus of 243 GPa and yield stress of 20 GPa[63]).

For the high-stress shots (>400 GPa), the diamond window free-surface velocity, $u_{fs}(t)$, was recorded as a function of time (Fig. 4B, blue and red traces) and used with a backward characteristics analysis to convert to stress using diamond ramp EOS[64-66]. The stress in SiC during the X-ray probe time was determined by spatially averaging over the sample layer (Fig. 4C, D). The stress uncertainty includes the following sources:[55] [1] uncertainty in the equation of state of the window materials (3% of the total stress[64,67], systematic); [2] uncertainty in determining the phase of the VISAR fringes (1% of the total stress, random); [3] uncertainty in the stress gradient over the X-ray probe timing. This is estimated as the standard deviation of the stress histogram fitted with a Gaussian function (random); [4] thickness uncertainty for each component of the target assembly (0.05% of the total stress, random); [5] uncertainty in the strength of diamond. Assuming the stress follows along the reversible isentropic path, maintaining diamond strength, leads to a systematic underestimate of the stress (systematic, +50 GPa[23]). Total stress uncertainties are calculated using sources [1]-[4] in quadrature. The additional +50 GPa uncertainty applies only to the shots with diamond windows [5]. The total stress uncertainties are shown in Supplementary table 2. A summary of laser power, stress and sample velocity as a function of time for all shots is shown in Supplementary Fig. 5.

**Calculation of principal isentrope and isotherm.** In these experiments, the longitudinal stress, $P_x$, was measured. The longitudinal stress-density, $P_x$-$\rho$, relationship was transformed to conditions of the principal isentrope by correcting for the following factors: (1) the deviation between the longitudinal stress and hydrostatic pressure, (2) the thermal pressure of the initial shock (~52 GPa) from a

porous starting material (B3 phase), (3) plastic work heating resulting from the strength of SiC and (4) the energy associated with the B3 to B1 phase transition at 100 GPa[15,17].

1. The longitudinal stress can be described as a sum of a hydrostatic pressure, $P_{hyd}$ and a stress deviator term. Using the von Mises yield criterion, the longitudinal stress is defined as

$$P_x = P_{hyd} + \frac{2}{3}Y, \tag{1}$$

where $Y$ is the yield strength.

2. The pressure-density relationship along the Hugoniot including porosity is given by:

$$P_H^* = P_H \frac{1 - \left(\frac{\gamma}{2}\right)\left(\frac{\rho}{\rho_0} - 1\right)}{1 - \left(\frac{\gamma}{2}\right)\left(\frac{\rho}{\rho_0^*} - 1\right)}, \tag{2}$$

where $P_H^*$ and $P_H$ are the stress states corresponding to the porous Hugoniot and to the crystal Hugoniot, respectively. $\rho_0$ and $\rho_0^*$ are the initial crystal and the initial bulk density, respectively. $\gamma$ is the Grüneisen parameter, assumed to be dependent only on density as given by:

$$\gamma = \gamma_0 \left(\frac{\rho_0}{\rho}\right)^q. \tag{3}$$

where $q$ is the logarithmic density dependence of $\gamma$. $\gamma_0$ is assumed to be in the range of 0.5-2.0 based on previous studies[17,18,63]. The laser drive in our experiments produces an initial elastic shock wave (~81 GPa) in the diamond ablator[64]. This transmits a shock with lower amplitude (~52 GPa) into the SiC sample due to the impedance mismatch between diamond and SiC. The thermal pressure between the principal isentrope and the Hugoniot at the conditions of the initial shock (~52 GPa) was determined using the Mie-Grüneisen relation[68],

$$P_H^* - P_S = \gamma\rho\left(E_H^* - E_S\right), \tag{4}$$

where $E_H^*$ is the energy along the porous Hugoniot, obtained from the Rankine-Hugoniot relationship:

$$E_H^* = \frac{P_H^*}{2}\left(\frac{1}{\rho_0^*} - \frac{1}{\rho}\right). \tag{5}$$

The pressure-density relationship of B3 SiC along the principal isentrope up to 52 GPa, $P_S$, was determined using the Birch-Murnaghan equation of state:

$$P_S = 3K_{OS}f\left(1 + 2f\right)^{\frac{5}{2}}\left(1 + \frac{3}{2}\left[K_{0S}' - 4\right]f\right), \tag{6}$$

where $f$ is the Eulerian strain:

$$f = \frac{1}{2}\left[\left(\frac{\rho}{\rho_0}\right)^{\frac{2}{3}} - 1\right], \tag{7}$$

and $K_{OS}$ and $K_{0S}'$ are the ambient isentropic bulk modulus and its pressure derivative (at constant entropy), respectively. The internal energy along the isentrope of the B3 phase up to 52 GPa, $E_S$, can be obtained by integration of the Birch-Murnaghan equation:

$$E_S = \int_{1/\rho_0}^{1/\rho} \frac{P_S}{\rho^2}d\rho = \frac{9}{2}\frac{K_{0S}}{\rho_0}\left(f^2 + \left[K_S' - 4\right]f^3\right). \tag{8}$$

3. Material strength under compression results in plastic work heating, a source of thermal pressure, which causes the hydrostatic pressure to deviate from the isentrope by

$$P_{hyd} - P_S = \gamma\rho\int_0^{\epsilon_x} f_{TQ}dW_P, \tag{9}$$

where $\epsilon_x$ is the natural strain $\ln\frac{\rho}{\rho_0}$, $f_{TQ}$ is the Taylor-Quinney factor, which describes the fraction of plastic work converted to heat[69], $W_p$ is the plastic working heating. The Taylor-Quinney factor represents the relationship between plastic work and heat generation, which is generally assumed to be $0.7 < f_{TQ} < 0.95$[70]. In this work, $f_{TQ}$ is assumed to be 1 indicative of 100% of plastic work converted to heat. Plastic working heating[71] is defined as

$$W_P = \frac{1}{\rho_0}\frac{2}{3}Y\left[d\epsilon_x - \frac{dY}{2G(\rho)}\right], \tag{10}$$

where $G(\rho)$ is the shear modulus. In this work, we assume $Y$ is fixed at a constant value ranging from 5- 20 GPa.

4. $E_T$ is the energy associated with the B3 to B1 phase transition[27]. This transition has a negative Clapeyron slope[18,20], indicating heat is absorbed (endothermic) which decreases the temperature across the transition.

The principal isentrope is calculated from the measured longitudinal stress through 1-4 given by

$$P_S = P_x - \frac{2}{3}Y - \gamma\rho\left(E_H^* - E_S\right) - \gamma\rho\int_0^{\epsilon_x}\beta dW_P + \gamma\rho E_T. \tag{11}$$

The 300-K isotherm is then determined from the principal isentrope using the Mie-Grüneisen relation. The thermodynamic parameters used in the calculation are listed in the Supplemental Material, Table 4.

**Stress, energy, and temperature path**. While ramp compression reduces the extent of sample heating, irreversible work and heating as a result of dissipation and material strength occur during ramp loading. The quasi-isentropic path followed in a ramp compression experiment lies between the Hugoniot and the isentrope (Supplementary Figures 6 and 7). In these experiments, the measurements of $P_x$-$\rho$ provide the energy along the ramp, $E_r$. using Eq. (11) and the Mie-Grüneisen relation,

$$E_r = E_S + \frac{2Y}{3\gamma\rho} + \left(E_H^* - E_{H,S}\right) + \int_0^{\epsilon_x} \beta \frac{1}{\rho_0} \frac{2}{3} Y d\epsilon_x - \left(E_{Tr}\right). \quad (12)$$

Below the B3-B1 transition pressure transition (<100 GPa), $E_r$ is divided into two regions: (1) the energy along the B3 porous Hugoniot up to 52 GPa, and (2) the energy along the isentrope of the B3 phase from the 52-GPa Hugoniot state to 100 GPa (Supplementary Figure 6). Above the B3-B1 phase transition, the estimated values of $E_r$ as a function of stress are calculated using Eq. 12 and are shown as the light-blue band in Supplementary Fig. 6.

Temperature cannot be directly measured in our experiments, but can be estimated from the energy difference between the ramp compression path and the principal isentrope, calculated using the Debye model given by:

$$E_r - E_S = 9nRT \left[\frac{\theta_D(V)}{T}\right]^{-3} \int_0^{\theta_D(V)/T} \frac{x^3}{e^x - 1} dx, \quad (13)$$

where $\theta_D(V)$ is the Debye temperature, $n$ is the number of atoms per formula unit, $R$ is the gas constant. The estimated temperature are shown as a blue curve in Supplementary Figure 7 assuming Y and $\gamma_0$ are 20 GPa and 1, respectively, corresponding to the $E_r$ shown as a black curve in Supplementary Figure 6. The estimated temperature ranges from 1160 K to 3350 K corresponding to a stress range from 100 GPa to 1600 GPa, which is much lower than the expected melting curve at these conditions shown in Supplementary Fig. 7.

## Data availability

Raw data in this study were generated at the Omega laser facility. The processed pulse shapes, VISAR, and the X-ray diffraction data are archived in the Department of Geosciences community of Princeton University's DataSpace: https://doi.org/10.34770/60th-e253.

## Code availability

Igor scripts used to analyze the X-ray diffraction and VISAR data described in here can be obtained from the corresponding author upon reasonable request.

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

## Acknowledgements

We thank Carol Davis (LLNL) and the staff of the Laboratory for Laser Energetics for expert experimental assistance. The Lawrence Livermore National Laboratory Analyze-VISAR and AnalyzePXRDIP codes were used for data analysis. The authors acknowledge the use of Princeton's Imaging and Analysis Center and Bolton Howes for assistance with porosity estimation. Portions of this work were performed under the auspices of the US Department of Energy by Lawrence Livermore National Laboratory under contract number DE-AC52-07NA27344. The research and materials incorporated in this work were partially developed at the National Laser Users' Facility at the University of Rochester's Laboratory for Laser Energetics, with financial support from the U.S Department of Energy under Cooperative Agreement DE-NA0001944 and DE-NA0003611. Additional support was provided by the National Science Foundation EAR-1644614 and the Princeton Center for Complex Materials (PCCM), and a National Science Foundation (NSF)-MRSZECZ program (DRM-2011750).

## Author contributions

D.K, R.F.S., J.H.E., S.J.T. and T.S.D conceived and designed the experiments. D.K. analyzed the data with support from R.F.S and F.C. D.K., R.F.S., I.K.O., M.C.M., M.K.G., J.K.W., J.R.R., and T.S.D. carried out experiments. J.R.R., A.L., F.C., M.M., and R.F.S. provided experimental development. D.K. and T.S.D wrote the manuscript with support from all authors on the manuscript. T.S.D. supervised the project. All authors were involved in discussions related to experimental design and data analysis.

## Competing interests

The authors declare that they have no competing interests.
