## [Peer Review File · Nature Communications]

REVIEWER COMMENTS

Reviewer #1 (Remarks to the Author):

This paper presents the phase determination and equation of state of shocked silicon carbide between 80 and 1500 GPa using ramp compression and in situ X-ray diffraction techniques. The authors then use their equation of state data to compute theoretical mass-radius curves of Si- and C-rich planets. This paper is excellent and I recommend it for publication in Nature Communications. The paper is clearly written and the techniques used are cutting edge. This study greatly expands our understanding of an important planetary and ceramic material over a very wide range of conditions.

I have a few comments that could increase clarity. They are detailed below.

Line 82: I noticed in Figure 5 and some of the supplemental figures that the data become really scattered around 400 GPa. Somewhere in here the authors could point out that having only one diffraction peak is likely a major reason behind the scatter.

Line 94-95 ("It should be noted..."): I think having this statement does the authors a disservice. Here they could make clear that it is a single datapoint (not a range of data at 433 GPa) and just point to the discussion in the supplement. Or just remove the sentence here and do the directing in the figure caption.

Line 198-202: Did the authors characterize their starting material using XRD as well? How did they get ambient pressure density of the pressed material?

Figure 3: The authors could explicitly say that the horizontal lines represent the material layers, as they did in Figure 4.

Figure 5 caption: I think the authors should explicitly point out in the caption the one datapoint that was left out of the fitting. That way they can explain the parentheses in the figure.

Table S2 caption: I believe there is an error here. ..."corresponding to a density of 7.70(24) g/cm³" should be 7.70(28) g/cm³. It's a very small thing, but I noticed it.

Table S4 header: The authors should specify which material these parameters correspond to in the header. Also, I think it would be more grammatically correct to change "low-pressure phases of SiC due to those..." to "low-pressure phases of SiC since those of B1-SiC are unknown".

Table S5: Are the fixed parameters assumed based on similar materials? It is probably safest to say where they came from.

Reviewer #2 (Remarks to the Author):

Review of "Structure and Density of Silicon Carbide to 1.5 TPa: Implications for Extrasolar Planets" by Kim and co-authors.

The paper presents new experimental data on silicon carbide, collected during laser driven ramp compression experiments. The obtained data are then used to model mass radius evolution for two ideal carbon enriched exoplanets.

While the extension of the investigated pressure range is truly impressive, I am not sure that in its current form makes it suitable for publication on Nature Communication. The experimental part is clear and detailed, but the exoplanets frame should be improved and integrated with further details and descriptions. Some logical links necessary to understand the topic are

missing, and it should be made clear to the reader where we stand in the topic, what we know and what we can say. In its current form the paper seems to suggest that there might be planets made entirely by SiC. This is not the case but is still important to study it because it provides references to then move to more complex systems. This should be made clear, otherwise a reader not familiar with the topic might be extremely misled.

I will report below some of the comments that I would like to see addressed before considering suggesting the paper for publication and some observations that I hope will help the author improving the scientific case.

1) Temperature: I understand that dealing with temperature in dynamic compression experiments is not trivial. It is estimated in the experimental methods but then not used in the manuscript where only a 300K EoS seems to be used.

I am questioning the validity of using a standard ambient temperature EoS to fit data that are probably also under the effect of temperature, is it reliable and representative of the dataset? And how valid is the comparison with the data from static compression experiments? Some might argue that the discrepancies in the evolution of the equation of states come from the use of ambient temperature EoS versus Thermal EoS. These data agree with Kidokoro et al., also using ambient temperature EoS, and doesn't agree with Miozzi et al, that account for temperature effects with a fitted thermal model. Maybe if the latter has ambient temperature data those would need to be use for the comparison instead of the thermal model?

I realize that using ambient temperature EoS is probably the way in which things need to be done with this type of data, but if it is, it should be addressed, because seeing it compared with a thermal one, and then used to make mass radius modelling, neglecting thermal contribution, is a bit puzzling.

Furthermore, data coming from experiments are often used for models by researchers in the exoplanet's community. The limitations and details, especially for such a valuable dataset should be made well clear to be sure that it will not be "improperly" used by people not familiar with this type of experiments.

2) This comment is related to the previous one, how reliable is a mass radius plot calculated without considering thermal effects?

How is the B3-B1 transition considered in the models?

Line 23: The paper is focused on exoplanets and the material applications are not detailed or mentioned again. I think that the last sentence of the abstract can be removed.

Lines 41-42: There is a logical jump from composition of the stars to details on what has been done on SiC. How do we know how which are the candidate materials? There are key words that need to be there, as for example condensation sequence. With all the works that have already been published on the topic the introduction can't be limited to the C/O ratio. Not for a paper that made making the planetary application one of the focal points.

Lines 120-130:. Analyses have been done on how light elements incorporation and the chosen core mantle fraction affect M/R plots (see the works from Unterborn et al. 2016 or Dorn et al. 2015 for example). How your results relate with those? I think it should be mentioned in the discussion.

Lines 133-137: The layered convection was proposed by Nisr et al. because of the properties of SiC. However, the dynamic of an SiC mantle has been numerically modelled in Miozzi et al. and the results show how it would be unlikely to have a layered convection. Please, before proposing hypothesis provide some references to what has already been done, for the reader to have a complete view.

This comment also applies to lines 138-144 where the proposed hypothesis has in part already been tested and modelled.

Line 162: "...would be less dense than a corresponding rocky planet", I think that a carbon enriched planet would be still considered rocky planet. A more suitable definition would probably be a silicate-dominated planet, or an Earth-like planet.

Line 174-175: "Planetary accretion simulations suggest carbon-rich planets may be dominated by a small number of components, i.e. SiC, C, and Fe¹". Please rephrase, this sentence is misleading, while Bond et al might not specify, condensations sequences showed in Madhusudan et al. 2012 and reported in Duffy et al.2015 show that iron carbides will condensate even before Fe. It should be made clear to a reader not familiar with the carbon rich exoplanets topic, that considering only SiC, C, Fe and FeSi, is an underestimation of the complexity of those planets, useful, because it provides a reference, but still an underestimation.

Line 188: "...enable the first experimentally constrained mass-radius curves". Please rephrase, I understand what you mean but the sentence is misleading. This is not the first experimentally constrained mass radius curve, it's the first time there is no need to extrapolate.

Reviewer #3 (Remarks to the Author):

The manuscript reports ramp compression experiments on silicon carbide (SiC) up to 1507 GPa. Based upon X-ray diffraction measurements, the authors inferred several solid-solid phase transitions. They fit their data to several equations of state then used to comment on possible exo-planet interiors.

The noteworthy results here are the ramp compression of an important and probably fairly common planetary material up to pressures expected to occur in massive terrestrial planetary interiors. All the data is original, although the methods employed in this study are not. I suspect that the EOS will be valuable for future planetary science studies. On the other hand, basically any material is useful for planetary science studies now because astronomers have found an incredible diversity of planets. Given that the methods employed here are, while not quite common yet, also not exactly novel, and that the measurements did not provide complete thermodynamic information about this material, the paper in its current form might be better suited for a more specialized journal.

That said, I think the manuscript could be strengthened, and its appeal broadened if the authors could address the following points:

1.) In a typical ramp experiment, the main observable is the time history of the particle velocity, or equivalently, the sound speed. It seems that this data was not presented. I guess it could be derived from panel C of fig. 4. But owing to its importance, the omission stuck out to me. Are you doing something different in this experiment which prevents you from measuring it?

2.) Powder was used, but it looks like the EOS model derived from the data assumed single crystal V_0 (reference volume). Was the density of the actual samples that were used in these experiments measured? If so, were they free of voidspace?

3.) Both the Vinet and Birch-Murnaghan EOS's assume isothermal compression, but the experimental data are (quasi)isentropic. What, if any, corrections were applied to the data used in the EOS fit? The authors assumed a thermal model for this material in their hydro calculations which was used to inform and

constrain their experimental data, so it should be easy for the authors to estimate the thermal contribution to their estimated pressures. This has been done in several previous papers by several of the co-authors. E.g. Smith et al. Nat. Astro. 2, (2018), and Fratanduono et al. Science 372, (2021), to name but a few. It seems to me that this is a minimum requirement for the work to be published.

4.) There are many assumptions about the response of the material that have been invoked in order to produce the dataset in this manuscript. It is commendable that the authors have so clearly stated exactly how the sausage was made. On the other hand, it's somewhat unappealing that these experiments depend so intimately on pre-existing models. Does X-ray diffraction loosen the coupling between data and modeling for these sorts of experiments? If so, this would seem to be an important point to impress upon the reader.

5.) If, as is stated in the introduction, carbon planets are expected to have highly exotic interiors, is the neglect of temperature in the construction of the planetary interior model justified? For more conventional compositions, ignoring temperature and the crust is probably fine so long as the astronomers can't measure planetary radii to better than a few percent. And they assumed a core that is proportional to that of what one might expect for a Earth-like terrestrial planet. But, having done that, what follows is essentially pure speculation. On the other hand, it would be quite interesting for the authors to examine some of these hypothesis using an improved planetary interior model derived from their data.

6.) Similarly, if you carried out the same planetary model construction on the old EOS models, is there some significant change? In other words, what have these new measurements done to improve our understanding of possible planetary interiors which we could not have inferred from previous studies?

The changes made to the manuscript during revision have been highlighted in the main text in yellow and supplementary material. We address specific comments from the reviewers below.

Reviewer #1 (Remarks to the Author):

This paper presents the phase determination and equation of state of shocked silicon carbide between 80 and 1500 GPa using ramp compression and in situ X-ray diffraction techniques. The authors then use their equation of state data to compute theoretical mass-radius curves of Si- and C-rich planets. This paper is excellent and I recommend it for publication in Nature Communications. The paper is clearly written and the techniques used are cutting edge. This study greatly expands our understanding of an important planetary and ceramic material over a very wide range of conditions.

We would like to thank the reviewer again for providing valuable comments on the manuscript.

I have a few comments that could increase clarity. They are detailed below.

Line 82: I noticed in Figure 5 and some of the supplemental figures that the data become really scattered around 400 GPa. Somewhere in here the authors could point out that having only one diffraction peak is likely a major reason behind the scatter.

The reviewer is correct. The data become scatter above 400 GPa, and this is a well-known feature of these experiments. The primary cause is that the X-ray background produced by the drive plasma strongly increases in intensity and energy as the laser power is increased to produce higher stresses. For this reason, the overall SNR is reduced and weaker diffraction lines can no longer be observed. We have added text in lines 87-88 to address this.

Line 94-95 (“It should be noted...”): I think having this statement does the authors a disservice. Here they could make clear that it is a single datapoint (not a range of data at 433 GPa) and just point to the discussion in the supplement. Or just remove the sentence here and do the directing in the figure caption.

We agree with the reviewer and have moved the discussion on the figure caption and the footnote to suppl. Table 2.

Line 198-202: Did the authors characterize their starting material using XRD as well? How did they get ambient pressure density of the pressed material?

The starting material was characterized by Raman spectroscopy which showed good consistency with literature values for B3 SiC.

The cold-pressed sample is too small and fragile to directly measure its bulk density. To address the porosity question, we obtained a scanning electron microscope (SEM) image from a sample prepared in an identical way to those used in our experiments. Using the image analysis method of Howes et al. (2021) [Ref. 55], the porosity was estimated to be small, ~2%.

The only effect that this porosity has on our experiments is to modestly increase the amount of heating due to the initial shock wave. We have added text in lines 234-237 about this. The representative data are shown in Fig. S4 in the supplemental material. The effects of porosity are incorporated into our energy and temperature calculations in Figs S6,7.

Figure 3: The authors could explicitly say that the horizontal lines represent the material layers, as they did in Figure 4.

We added sentences in Figs. 3 and 4.

Figure 5 caption: I think the authors should explicitly point out in the caption the one datapoint that was left out of the fitting. That way they can explain the parentheses in the figure.

We added the sentences in the caption in Figure 5.

Table S2 caption: I believe there is an error here. ...“corresponding to a density of 7.70(24) g/cm³” should be 7.70(28) g/cm³. It’s a very small thing, but I noticed it.

Thank you for the careful reading. The error was corrected.

Table S4 header: The authors should specify which material these parameters correspond to in the header. Also, I think it would be more grammatically correct to change “low-pressure phases of SiC due to those...” to “low-pressure phases of SiC since those of B1-SiC are unknown”.

Corrected.

Table S5: Are the fixed parameters assumed based on similar materials? It is probably safest to say where they came from.

There are few constraints on the Grüneisen parameter and its volume dependence for the B1 phase of SiC, and existing constraints are limited to low pressure (<200 GPa). We expect our work will stimulate further theoretical and experimental studies to better constrain the thermoelastic properties of this material over a wide pressure range. We assumed γ_0 is in the range of 0.5-2.0 to calculate a theoretical B1-SiC Hugoniot using a Mie-Grüneisen EOS. This is in the range reported by the available literature (Varshney et al 2015; Miozzi et al. 2018; Tracy et al. 2019). It is very common to assume $q=1$ in dynamic compression studies. The

results do not change greatly if we use the parameters of Miozzi et al ($\gamma_0=0.5$, $q=1.6$)

Reviewer #2 (Remarks to the Author):

Review of “Structure and Density of Silicon Carbide to 1.5 TPa: Implications for Extrasolar Planets” by Kim and co-authors.

The paper presents new experimental data on silicon carbide, collected during laser driven ramp compression experiments. The obtained data are then used to model mass radius evolution for two ideal carbon enriched exoplanets.

While the extension of the investigated pressure range is truly impressive, I am not sure that in its current form makes it suitable for publication on Nature Communication. The experimental part is clear and detailed, but the exoplanets frame should be improved and integrated with further details and descriptions. Some logical links necessary to understand the topic are missing, and it should be made clear to the reader where we stand in the topic, what we know and what we can say. In its current form the paper seems to suggest that there might be planets made entirely by SiC. This is not the case but is still important to study it because it provides references to then move to more complex systems. This should be made clear, otherwise a reader not familiar with the topic might be extremely misled.

I will report below some of the comments that I would like to see addressed before considering suggesting the paper for publication and some observations that I hope will help the author improving the scientific case.

We would like to thank the reviewer again for providing valuable comments on the manuscript. As described below, we have provided further details and descriptions of the exoplanet discussion. We have stated that the model of a pure SiC planet is a hypothetical end-member used to explore the possible compositional space of extrasolar planets, as has been done for other candidate materials in the literature. We have modified the text to make this more clear.

1) Temperature: I understand that dealing with temperature in dynamic compression experiments is not trivial. It is estimated in the experimental methods but then not used in the manuscript where only a 300K EoS seems to be used.

I am questioning the validity of using a standard ambient temperature EoS to fit data that are probably also under the effect of temperature, is it reliable and representative of the dataset? And how valid is the comparison with the data from static compression experiments? Some might argue that the discrepancies in the evolution of the equation of states come from the use of ambient temperature EoS versus Thermal EoS. These data agree with Kidokoro et al., also using an ambient temperature EoS, and doesn't agree with Miozzi et al, that account for temperature effects with a fitted thermal model. Maybe if the latter has ambient temperature data those would need to be used for the comparison instead of the thermal model?

In order to address the reviewer's comment, we have extended our calculations to enable us to reduce our data to an isentrope and isotherm. We added text at lines 89-105 and 326-405.

First we calculate the energy increase due to the initial shock (including the effect of sample porosity). Then we can calculate the additional energy increase due to ramp compression to the final state by taking into account the effects of plastic work and the phase transition energy. This requires assumptions about the yield strength and thermodynamic parameters which are not well constrained for the B1 phase of SiC. We can then use the Mie-Grüneisen equation to calculate the change in energy from the ramp state to the principal isentrope and further reduce to an isotherm using a Debye model.

The correction from the conditions of our experiment to the isotherm or isentrope is small and does not change our conclusions in any way. Our work is the first to explore this material in the solid state over this wide and very high pressure range and further work will be needed to better constrain the strength and other properties at these conditions.

I realize that using ambient temperature EoS is probably the way in which things need to be done with this type of data, but if it is, it should be addressed, because seeing it compared with a thermal one, and then used to make mass radius modelling, neglecting thermal contribution, is a bit puzzling. Furthermore, data coming from experiments are often used for models by researchers in the exoplanet's community. The limitations and details, especially for such a valuable dataset should be made well clear to be sure that it will not be "improperly" used by people not familiar with this type of experiments.

We have corrected the data to an isentrope as discussed above. For the thermodynamic parameters used, there is little difference between an EOS fit to the ramp data and the reduced isentrope. Furthermore, the difference between pressures along the isentrope and isotherm is very small at high level of compression in our work as well as Ref. 73. This does not change the conclusions of our paper.

2) This comment is related to the previous one, how reliable is a mass radius plot calculated without considering thermal effects?
How is the B3-B1 transition considered in the models?

The effects of pressure, composition, and phase transitions are much larger than the effects of temperature in modeling large, rocky exoplanets. Temperature effects on the equation of state for rocky exoplanets were evaluated by Seager et al. (2007) and found to increase the radius of planets by only 1.2 %. As a result, it is common to not explicitly include temperature in modeling these planets. In the case of SiC, temperature is likely even less important as the thermal expansivity is very low ($5.6 \times 10^6 \text{ K}^{-1}$ for B1 phase from Miozzi et al. (2018)). Pressure further decreases thermal expansivity reducing the importance of temperature effects at the very high pressure conditions of rocky exoplanets.

An advantage of our work is that it is the first to incorporate the B3-B1 transition in models of carbon-rich rocky planets. We fixed the transition pressure at 70 GPa (Miozzi et al., 2018 and Kidokoro et al. 2017) which corresponds to depths of ~ 800-km to 2500-km for carbon-rich

planet with 1-10 M_E . Incorporation of this low-density region will affect the mass-radius relationship. Without taking into account for B3 phase, the radius of planets are found to be ~1% to 5% smaller than corresponding carbon-rich planet with 1-10 Earth masses. Thus, it is a factor that should be considered, especially for smaller planets. Furthermore, the large density change across the transition may have strong dynamical consequences for the planet as we discuss in the manuscript.

3) Line 23: The paper is focused on exoplanets and the material applications are not detailed or mentioned again. I think that the last sentence of the abstract can be removed.

We have deleted the last sentence.

4) Lines 41-42: There is a logical jump from composition of the stars to details on what has been done on SiC. How do we know how which are the candidate materials? There are key words that need to be there, as for example condensation sequence. With all the works that have already been published on the topic the introduction can't be limited to the C/O ratio. Not for a paper that made making the planetary application one of the focal points.

We expanded this discussion in lines 27-29.

5) Lines 120-130: Analyses have been done on how light elements incorporation and the chosen core mantle fraction affect M/R plots (see the works from Unterborn et al. 2016 or Dorn et al. 2015 for example). How your results relate with those? I think it should be mentioned in the discussion.

Good point. We added some discussion regarding the possible presence of light elements and their effects in lines 126-128, 143-144, and core mantle fraction compared with a Earth-like planet in lines 195-198.

6) Lines 133-137: The layered convection was proposed by Nisr et al. because of the properties of SiC. However, the dynamic of an SiC mantle has been numerically modelled in Miozzi et al. and the results show how it would be unlikely to have a layered convection. Please, before proposing hypothesis provide some references to what has already been done, for the reader to have a complete view. This comment also applies to lines 138-144 where the proposed hypothesis has in part already been tested and modelled.

We added text at lines 149-162 and 173-176 to address the previous literature.

7) Line 162: "...would be less dense than a corresponding rocky planet", I think that a carbon enriched planet would be still considered rocky planet. A more suitable definition would probably be a silicate-dominated planet, or an Earth-like planet.

Thank you for pointing out. We have modified the text to read as follows: "a corresponding Earth-like planet" in line 194.

8) Line 174-175: “Planetary accretion simulations suggest carbon-rich planets may be dominated by a small number of components, i.e. SiC, C, and Fe1”. Please rephrase, this sentence is misleading, while Bond et al might not specify, condensations sequences showed in Madhusudan et al. 2012 and reported in Duffy et al.2015 show that iron carbides will condensate even before Fe. It should be made clear to a reader not familiar with the carbon rich exoplanets topic, that considering only SiC, C, Fe and FeSi, is an underestimation of the complexity of those planets, useful, because it provides a reference, but still an underestimation.

We agree and have modified the text in lines 118-119.

9) Line 188: “...enable the first experimentally constrained mass-radius curves”. Please rephrase, I understand what you mean but the sentence is misleading. This is not the first experimentally constrained mass radius curve, it’s the first time there is no need to extrapolate.

We agree and have modified the text in lines 220-222.

Reviewer #3 (Remarks to the Author):

The manuscript reports ramp compression experiments on silicon carbide (SiC) up to 1507 GPa. Based upon X-ray diffraction measurements, the authors inferred several solid-solid phase transitions. They fit their data to several equations of state then used to comment on possible exo-planet interiors.

The noteworthy results here are the ramp compression of an important and probably fairly common planetary material up to pressures expected to occur in massive terrestrial planetary interiors. All the data is original, although the methods employed in this study are not. I suspect that the EOS will be valuable for future planetary science studies. On the other hand, basically any material is useful for planetary science studies now because astronomers have found an incredible diversity of planets. Given that the methods employed here are, while not quite common yet, also not exactly novel, and that the measurements did not provide complete thermodynamic information about this material, the paper in its current form might be better suited for a more specialized journal.

We thank the reviewer for his/her helpful comments, and for noting that our study will be “valuable for future planetary science studies.” We agree that many materials have planetary applications because of the large variety of compositions expected in extrasolar planets, and SiC is key to understand carbon-rich planets. Nonetheless, relatively few studies have been performed on SiC, compared with metals and silicates, especially at the extreme conditions expected within large exoplanets. We therefore believe that our characterization of the structure and equation of state of SiC at TPa pressure is extremely important for the modeling the interiors of such planets. Furthermore, most previous studies using these techniques have been performed on dense metals and our work is one of a very few to show that a low-density

ceramic material can be studied at conditions above 1 TPa. SiC has long been of great interest under static and dynamic high pressure, as there is a very large literature on these topics and extending the examined pressure range by a factor of 7 is a significant step forward.

That said, I think the manuscript could be strengthened, and its appeal broadened if the authors could address the following points:

1.) In a typical ramp experiment, the main observable is the time history of the particle velocity, or equivalently, the sound speed. It seems that this data was not presented. I guess it could be derived from panel C of fig. 4. But owing to its importance, the omission stuck out to me. Are you doing something different in this experiment which prevents you from measuring it?

In ramp experiments with x-ray diffraction, the sound velocity is not measured. Sound speed measurements under ramp loading require stepped targets with different thicknesses to measure wave arrival times at different distances (e.g., Bradley et al. 09, Smith et al, 2018). In XRD experiments, the free-surface or interfaces velocities are measured only at a single location and stress is determined from this using the methods described in the text.

2.) Powder was used, but it looks like the EOS model derived from the data assumed single crystal V_0 (reference volume). Was the density of the actual samples that were used in these experiments measured? If so, were they free of void space?

The porosity of the starting material was addressed in the comment to reviewer 1 above. The V_0 value for our equation of state is for the B1 phase, not the B3 phase of the starting material. The B1 is not quenchable so its volume cannot be directly measured. We used the value from the B1-EOS 300-K static-compression experiments by Miozzi et al (2018).

3.) Both the Vinet and Birch-Murnaghan EOS's assume isothermal compression, but the experimental data are (quasi)isentropic. What, if any, corrections were applied to the data used in the EOS fit? The authors assumed a thermal model for this material in their hydro calculations which was used to inform and constrain their experimental data, so it should be easy for the authors to estimate the thermal contribution to their estimated pressures. This has been done in several previous papers by several of the co-authors. E.g. Smith et al. Nat. Astro. 2, (2018), and Fratanduono et al. Science 372, (2021), to name but a few. It seems to me that this is a minimum requirement for the work to be published.

We have added a section to the text on this. We made corrections for the effects of the initial shock including porosity, plastic work heating, deviatoric stress, and phase transition energy to estimate the principal isentrope and isotherm by making reasonable estimates of strength and thermodynamic properties. As discussed in the response to rev. 2, the resulting equation of state parameters change only slightly and our conclusions are not altered in any way.

4.) There are many assumptions about the response of the material that have been invoked in

order to produce the dataset in this manuscript. It is commendable that the authors have so clearly stated exactly how the sausage was made. On the other hand, it's somewhat unappealing that these experiments depend so intimately on pre-existing models. Does X-ray diffraction loosen the coupling between data and modeling for these sorts of experiments? If so, this would seem to be an important point to impress upon the reader.

We are not sure what the reviewer is referring to when they say that: “these experiments depend so intimately on pre-existing models”. The XRD data directly probes the properties of the sample and this provides the main result of our study. The stress is determined from velocimetry measurements that build on well-established earlier results which is common practice. We believe that it is important to provide sufficient details on the analysis procedure to make it clear to a broader audience who may not be familiar with dynamic compression experiments and data analysis.

5.) If, as is stated in the introduction, carbon planets are expected to have highly exotic interiors, is the neglect of temperature in the construction of the planetary interior model justified? For more conventional compositions, ignoring temperature and the crust is probably fine so long as the astronomers can't measure planetary radii to better than a few percent. And they assumed a core that is proportional to that of what one might expect for a Earth-like terrestrial planet. But, having done that, what follows is essentially pure speculation. On the other hand, it would be quite interesting for the authors to examine some of these hypothesis using an improved planetary interior model derived from their data.

This has been addressed in comment 2) from the second reviewer. Temperature is a second-order effect on the calculated radius of rocky exoplanets because thermal expansivity is very small at ultrahigh pressures.

6.) Similarly, if you carried out the same planetary model construction on the old EOS models, is there some significant change? In other words, what have these new measurements done to improve our understanding of possible planetary interiors which we could not have inferred from previous studies?

Existing models of the interiors of extrasolar planets are based on either long extrapolations of low-pressure data or theoretical calculations. The advantage of our study is that it provides an experimentally constrained EOS without need for extrapolation. In the low-pressure region, the differences between EOSs are negligible, but the discrepancies between them become pronounced above 300 GPa (see below the attached figure), which will affect the mass-radius relation for the large carbide planets. For instance, the difference in radius using the previous EOSs is ~3.3% for the hypothetical SiC planet with $10 M_E$ (see the figure 7 in the main text). Our experiments allow for not only resolving these discrepancies and providing accurate mass-radius relations which are useful to interpret the planet's interior compositions.

We would like to thank the reviewers again for a highly constructive discussion which has improved our manuscript. We hope that our paper is now acceptable for publication in *Nature Communications*.

Donghoon Kim
 Graduate student
 Department of Geosciences
 Princeton University

REVIEWERS' COMMENTS

Reviewer #3 (Remarks to the Author):

My main concern with the manuscript was in the construction of the EOS models. The authors have completely addressed my concerns and I see no further barriers to publication in Nature Communications.

Reviewer #3 (Remarks to the Author):

My main concern with the manuscript was in the construction of the EOS models. The authors have completely addressed my concerns and I see no further barriers to publication in *Nature Communications*.

We thank the reviewer for the positive comments and publication recommendation.

We would like to thank the reviewers again for a highly constructive discussion which has improved our manuscript. We hope that our current version complies with the journal formatting requirements and suitable for publication in *Nature Communications*.

Donghoon Kim
Graduate student
Department of Geosciences
Princeton University